# Hollow Mesoporous Silica Nanoparticles as a New Nanoscale Resistance Inducer for Fusarium Wilt Control: Size Effects and Mechanism of Action

**DOI:** 10.3390/ijms25084514

**Published:** 2024-04-20

**Authors:** Chaopu Ding, Yunfei Zhang, Chongbin Chen, Junfang Wang, Mingda Qin, Yu Gu, Shujing Zhang, Lanying Wang, Yanping Luo

**Affiliations:** School of Tropical Agriculture and Forestry, Hainan University, Haikou 570228, China; dingcp1227@163.com (C.D.); zhangyunfei2020@hainanu.edu.cn (Y.Z.); 21220951320049@hainanu.edu.cn (C.C.); 21220951320094@hainanu.edu.cn (J.W.); 21210904000018@hainanu.edu.cn (M.Q.); 20213007402@hainanu.edu.cn (Y.G.); 990992@hainanu.edu.cn (L.W.)

**Keywords:** hollow mesoporous silica nanoparticles, cowpea resistance, Fusarium wilt, salicylic acid, systemic acquired resistance

## Abstract

In agriculture, soil-borne fungal pathogens, especially *Fusarium oxysporum* strains, are posing a serious threat to efforts to achieve global food security. In the search for safer agrochemicals, silica nanoparticles (SiO_2_NPs) have recently been proposed as a new tool to alleviate pathogen damage including Fusarium wilt. Hollow mesoporous silica nanoparticles (HMSNs), a unique class of SiO_2_NPs, have been widely accepted as desirable carriers for pesticides. However, their roles in enhancing disease resistance in plants and the specific mechanism remain unknown. In this study, three sizes of HMSNs (19, 96, and 406 nm as HMSNs-19, HMSNs-96, and HMSNs-406, respectively) were synthesized and characterized to determine their effects on seed germination, seedling growth, and *Fusarium oxysporum* f. sp. *phaseoli* (FOP) suppression. The three HMSNs exhibited no side effects on cowpea seed germination and seedling growth at concentrations ranging from 100 to 1500 mg/L. The inhibitory effects of the three HMSNs on FOP mycelial growth were very weak, showing inhibition ratios of less than 20% even at 2000 mg/L. Foliar application of HMSNs, however, was demonstrated to reduce the FOP severity in cowpea roots in a size- and concentration-dependent manner. The three HMSNs at a low concentration of 100 mg/L, as well as HMSNs-19 at a high concentration of 1000 mg/L, were observed to have little effect on alleviating the disease incidence. HMSNs-406 were most effective at a concentration of 1000 mg/L, showing an up to 40.00% decline in the disease severity with significant growth-promoting effects on cowpea plants. Moreover, foliar application of HMSNs-406 (1000 mg/L) increased the salicylic acid (SA) content in cowpea roots by 4.3-fold, as well as the expression levels of SA marker genes of *PR-1* (by 1.97-fold) and *PR-5* (by 9.38-fold), and its receptor gene of *NPR-1* (by 1.62-fold), as compared with the FOP infected control plants. Meanwhile, another resistance-related gene of *PAL* was also upregulated by 8.54-fold. Three defense-responsive enzymes of POD, PAL, and PPO were also involved in the HMSNs-enhanced disease resistance in cowpea roots, with varying degrees of reduction in activity. These results provide substantial evidence that HMSNs exert their Fusarium wilt suppression in cowpea plants by activating SA-dependent SAR (systemic acquired resistance) responses rather than directly suppressing FOP growth. Overall, for the first time, our results indicate a new role of HMSNs as a potent resistance inducer to serve as a low-cost, highly efficient, safe and sustainable alternative for plant disease protection.

## 1. Introduction

Soil-borne fungal pathogens pose a vital threat to global food security causing up to 75% yield losses of major crops [1]. *Fusarium oxysporum* strains, a ubiquitous class of soil-borne pathogens ranking fifth in the top 10 plant pathogens of high importance, can infect hundreds of agricultural crop species, including legume crops [2]. For instance, Fusarium wilt of cowpea (*Vigna unguiculata*) is a severe disease caused by *Fusarium oxysporum* f. sp. *phaseoli* (FOP) and *Fusarium oxysporum* f. sp. *tracheiphilum* [3], which leads to annual yield losses of 30–100% around the world [4]. FOP was isolated as the preponderant species from diseased cowpea plants in China [4]. Chemical approaches and cultural practices have long been used to minimize the soil-borne disease outbreak. Unfortunately, neither of these strategies gives consistent and effective results for Fusarium wilt control with no detrimental effects on the environment and humans [1]. Thus, it seems urgent to develop new sustainable methods for effective control of Fusarium wilt of cowpea and many other crops.

Plants have evolved an innate immunity system that can be activated by both pathogen attack and resistance inducers to fend off potential pathogens. As an especially interesting form of induced resistance, systemic acquired resistance (SAR) is characterized by the spread of locally induced disease resistance to the whole plant [5]. SAR confers a long-lasting protection mechanism on pathogen-attacked plants. Salicylic acid (SA) is a key signal molecule in plants that is responsible for the activation of SAR via acting on the pathogen-related (PR) genes [5,6]. SAR can be initiated using resistance-inducing compounds without application of irreversible genetic modifications or fungicides with potential environmental risks [7]. This makes SAR a promising strategy for controlling soil-borne diseases, including Fusarium wilt [8,9,10].

Nano-agrochemicals have recently emerged as an alternative and complementary tool for conventional methods of chemical control, biological control, and cultural practices to improve crop yields and global food security, due to their advantages of better efficacy, reduced input, and lower ecotoxicity [11,12,13]. A wide variety of nanoparticles have been reported to control pathogen-infesting plants [14,15] including silica nanoparticles (SiO_2_NPs) [15], silver nanoparticles, and their nanocomposites [16]. SiO_2_NPs have many unique characteristics such as an adjustable pore structure, easy modification, simple synthesis, low cost, and less environmental harm [17]. Unlike other nanoparticles which exert their antimicrobial activities mainly by directly acting on the target pathogens, SiO_2_NPs exhibit diverse roles as an antimicrobial agent by directly restraining the virulence of plant pathogens [15,18,19], boosting plant innate immunity to alleviate pathogen damage [7,12,20,21,22], or delivering pesticides to enhance their efficacy [23]. Recently, as a controlled-release pool of bioavailable silicon (orthosilicic acid), SiO_2_NPs are beginning to attract significant attention as a new resistance inducer that can enhance disease resistance without negative effects on the crop growth and yield [7,9,12,20,21,22]. El-Shetehy et al. (2021) described the potential of SiO_2_NPs in enhancing the disease resistance of *Arabidopsis thaliana* against *Pseudomonas syringae* via the SA-dependent SAR pathway similar to conventional Si products [7]. Rice plants were also reported to employ SiO_2_NPs to decrease the incidence of *Magnaporthe oryzae* by the same manner [12]. For Fusarium wilt control, foliar spray of SiO_2_NPs significantly reduced the disease severity of *Fusarium oxysporum* f. sp. *niveum* in watermelons by modulating stress-related gene expression [9,10], and application of soil SiO_2_NPs also suppressed *Fusarium oxysporum* greatly in maize [24]. However, it is also not known whether SiO_2_NPs can enhance the disease suppression of Fusarium wilt in other crop plants. Of note, much still needs to be understood to elucidate the specific mechanism of SiO_2_NPs for control of Fusarium wilt.

Hollow mesoporous silica nanoparticles (HMSNs), a unique class of SiO_2_NPs with large hollow cavities and intact porous shells, have been accepted as desirable carriers for controlled pesticide delivery due to their robust and low-cost synthesis with tunable physicochemical properties and high loading capacity [25,26,27]. Previous studies have provided insights into the roles of solid core SiO_2_NPs in enhancing the disease resistance of several specific plants at a single diameter scale range [7,12,20,21,22]. It still remains unclear whether HMSNs can induce disease resistance in crop plants, and whether their performances are related to the diameter scales and SAR induction. To these ends, in this work, three spherical HMSNs of different diameters were synthesized and characterized. Using *V. unguiculata* as the target crop plant, we investigated the effects of three HMSNs on seed germination, seedling growth, and Fusarium wilt suppression through foliar treatments. HMSNs of optimal particle size and concentration were then determined and foliarly applied to cowpea to explore the supposed SAR pathway for Fusarium wilt control in terms of endogenous SA levels, resistance-related gene expression, and variations of defense-responsive enzyme activity. This is the first example known to the authors that reveals the role of HMSNs as a valuable plant immunity nano-inducer for Fusarium wilt control by means of the SA-dependent SAR pathway.

## 2. Results

### 2.1. Characterization of Hollow Mesoporous Silica Nanoparticles (HMSNs)

Three HMSNs of different sizes were synthesized to explore their roles in control of cowpea Fusarium wilt. Transmission electron microscopy (TEM) was employed to characterize their internal morphologies and size distributions. As shown in Figure 1a–c, representative HMSN morphologies of hollow cavities and intact porous shells were observed from the TEM images of the three HMSNs. TEM diameter analysis (Figure 1d–f) revealed that the three HMSNs had uniform hollow sphere structures with primary particle diameters of 19 ± 3 (HMSNs-19), 96 ± 9 (HMSNs-96), and 406 ± 26 nm (HMSNs-406). The surface morphologies of the three HMSNs were also observed by the scanning electron microscope (SEM) to characterize the nano spherical structures (Appendix A) consistent with the TEM observations. Hydrodynamic diameters and zeta potentials of the three HMSNs were also measured, with results showing larger sizes compared with the TEM data and negative surface potentials (Appendix A), consistent with previously reported results [7,9,10]. Characters of narrow particle size distributions of the three HMSNs (12–28, 70–125, and 350–455 nm, respectively), and their overall wide range scale (12–455 nm) that covers the existing application scope of SiO_2_NPs for plant disease control [7,9,10,12,15,20,21,22,28], make them desirable materials to elucidate the size effects of HMSNs for Fusarium wilt control.

### 2.2. Effect of HMSNs on Cowpea Seed Germination and Growth

Previous studies [10,12,19] have revealed that SiO_2_NPs are beneficial to seed germination and plant growth within appropriate concentration ranges. Thus, we first explored the effects of three synthesized HMSNs at different concentrations (100, 500, 1000, and 1500 mg/L) on seed germination and growth using a germination index (GI), and the root and shoot fresh biomass of seedlings as the indicators. As illustrated in Figure 2a–c, HMSNs-19 improved seed germination at 1000 mg/L with a higher GI value compared with the control, while the other two HMSNs, as well as HMSNs-19 at other concentrations, exhibited no significant effects on cowpea seed germination. Si(OH)_4_, however, showed a concentration-dependent inhibition of seed germination with declining GI values along with an increase in the concentration (Figure 2d). Compared with the control, seed germination was significantly inhibited when the concentration of Si(OH)_4_ exceeded 500 mg/L. In addition, the three HMSNs also exhibited no adverse impacts on the seedling growth, showing no significant differences in the root and shoot biomass at the test concentrations relative to the controls (Figure 3a–c). However, the root biomass was significantly decreased by Si(OH)_4_ at concentrations exceeding 500 mg/L and the shoot biomass was also significantly decreased by Si(OH)_4_ at a high concentration of 1000 or 1500 mg/L (Figure 3d). Overall, these results indicate that the three HMSNs pose no adverse effects on both the cowpea seed germination and seedling growth while Si(OH)_4_ can produce stress effects on both the seed germination and seedling growth at higher concentrations exceeding 500 or 1000 mg/L.

### 2.3. Effect of HMSNs on FOP Mycelial Growth

To test whether HMSNs have a direct toxic effect on *Fusarium oxysporum* f. sp. *phaseoli* (FOP) growth, the inhibitory effects of the three HMSNs on FOP mycelial growth were determined at concentrations ranging from 100 to 2000 mg/L in vitro, with results illustrated in Figure 4. Although FOP mycelial growth was to some extent inhibited by the three HMSNs at different concentrations, the inhibitory effects of the three HMSNs were very weak, showing inhibition ratios of less than 20% even at 2000 mg/L. Overall, none of the three HMSNs posed a strong inhibitory effect directly on FOP mycelial growth, allowing a further study to explore their performances for cowpea wilt control in vivo.

### 2.4. Effect of HMSNs on the Disease Incidence

The area under the disease progress curve (AUDPC) [9,10] was determined to evaluate the roles of HMSNs in suppressing Fusarium wilt of cowpea plants. As shown in Figure 5a, HMSNs exerted their benefits of disease suppression in a size- and concentration-dependent manner. HMSNs-19 were observed to have little effect on alleviating the disease incidence at both low and high doses with AUDPC values comparable to the infected control. The other two HMSNs, HMSNs-96 and HMSNs-406, exhibited similar capacities for significantly suppressing the disease incidence at the high concentration of 1000 mg/L, but scarcely working on the disease suppression at the low concentration of 100 mg/L. The disease progress was reduced by 23.13% and 40.00% for HMSNs-96 and HMSNs-406 treatments at 1000 mg/L (Figure 5a), respectively. Foliar application of SiO_2_NPs with a high concentration (1500 mg/L) was also previously described for the effective suppression of Fusarium wilt in watermelon [9]. Opposite effects, however, were observed for the Si(OH)_4_ treatments, with reduced disease incidence at the low concentration of 100 mg/L but a detrimental effect at a higher concentration of 1000 mg/L (Figure 5a). Similar concentration-dependent effects of Si(OH)_4_ were also observed on *Arabidopsis thaliana* infected by *Pseudomonas syringae* [7]. The difference of HMSNs and Si(OH)_4_ in the appropriate concentrations can be attributed to the previously described slow-release effect of HMSNs to generate Si(OH)_4_ instead of direct interaction [7].

Plant biomass was also monitored as another means of assessing the overall health of the plants under the diseased and healthy conditions, with results shown in Figure 5c. Cowpea growth was inhibited after being transplanted in the FOP infected soil for 29 days, with the fresh biomass of shoots and roots decreased by 14.63% and 20.78%, respectively. An overall trend of HMSNs was observed to promote the growth of cowpea plants, showing increased fresh biomass of shoots and roots relative to the infected control plants. The three HMSNs significantly enhanced the cowpea shoot and/or root biomass at a concentration of 100 mg/L as compared to the infected control, but did not affect the disease incidence. This similar character was also previously observed for sulfur nanoparticles in tomato Fusarium wilt control [29]. Of note, treatment with a concentration of 1000 mg/L of HMSNs-406 led to significant increases in both cowpea shoot and root biomass by 31.02% and 87.67%, respectively, relative to the infected controls. HMSNs-96 at a concentration of 1000 mg/L only significantly improved the fresh biomass of cowpea roots. Si(OH)_4_ of a high concentration of 1000 mg/L significantly lowered the fresh biomass of shoots, whereas its lower concentration of 100 mg/L improved the cowpea shoot and root biomass. This result may be attributed to high-dose Si(OH)_4_ inducing stress in the plant [7]. Phenotypic images of cowpea shoots, longitudinal-cut stems, and intact roots under HMSNs-406 (1000 mg/L) or Si(OH)_4_ (100 mg/L) treatment are presented in Figure 5b. Taken together, HMSNs-406 at a concentration of 1000 mg/L can be an optimal candidate for cowpea Fusarium wilt control due to both significant disease suppression and plant growth improvement.

### 2.5. Effect of Foliar HMSNs-406 Application on Cowpea Root Salicylic Acid (SA) Content

Since the findings that HMSNs-406 posed no obvious direct toxicity on FOP growth (Figure 4), whereas foliar spray of HMSNs-406 could significantly suppress the FOP incidence in the cowpea roots (Figure 5a), we consider this role as a kind of induced resistance of systemic acquired resistance (SAR) characterized by the spread of locally induced disease resistance to the whole plant [5,7,12]. The plant hormone SA plays a core regulatory role in plant SAR as a key signaling compound [5,7,12]. Recent findings have revealed that the control effect of SiO_2_NPs is involved in activation of SA-dependent plant immune responses called SAR [7,11]. Thus, we measured the SA content in cowpea roots to identify the primary pathway of SAR induction by HMSNs-406. As shown in Figure 6a, foliar application of HMSNs-406 (1000 mg/L) or Si(OH)_4_ (100 mg/L) increased SA content in cowpea roots by 4.3- and 3.4-fold, respectively, compared with the infected control. These findings indicate that the SAR activation in cowpea plants by HMSNs-406 or Si(OH)_4_ is clearly SA dependent. Although HMSNs-406 were demonstrated to trigger the production of SA and consequently increase the resistance of cowpea plants to FOP, it is unclear how SA production in cowpea roots was accelerated by HMSNs-406. Recently, using transcriptomic analysis, SiO_2_NPs have been found to enhance SA content through the regulation of SA-metabolism genes to confer the bacterial wilt resistance in peanuts while genes involved in SA biosynthesis were not affected [21]. Therefore, we speculate that HMSNs-406 might also act through regulating the expression of genes involved in SA biosynthesis or metabolism to result in the accumulation of SA, which remains to be verified.

### 2.6. Gene Expression Changes after Foliar Exposure of HMSNs-406

Pathogenesis-related (PR) genes including *PR-1* and *PR-5* are commonly used as markers for salicylic acid-mediated activation of SAR [7,30]. To further verify SA-dependent SAR induced by foliar treatment of HMSNs-406, we quantified the expression of *PR-1* and *PR-5* in FOP infected cowpea roots. Similar to treatment with Si(OH)_4_ (100 mg/L), treatment with HMSNs-406 (1000 mg/L) resulted in 1.97- and 9.38-fold increases in *PR-1* (Figure 6b) and *PR-5* (Figure 6c) in the transcript abundance, respectively, relative to the infected control. Non-expressed pathogen-related gene 1 (*NPR-1*) has been reported to be a receptor for salicylic acid and positively regulate the expression of *PR* genes [30]. Thus, *NPR-1* expression was also measured to clarify whether this upstream gene participated in the HMSNs-406-activated SAR. Foliar exposure of HMSNs-406 (1000 mg/L) boosted the expression level of *NPR-1* by 1.62-fold in the cowpea roots compared with the infected control, as did the Si(OH)_4_ (100 mg/L) by 1.76-fold (Figure 6d). Phenylalanine ammonia lyase gene (*PAL*), an antioxidant defense-related enzyme gene, is also associated with the response to pathogen-induced stress [29,31]. In cowpea infected roots, there was significant upregulation of *PAL* (8.54-fold) with HMSNs-406 foliar exposure (Appendix A), consistent with previously reported works [29,31]. These findings indicate that HMSNs-406, similar to Si(OH)_4_, can activate SA-dependent SAR defense in FOP infected cowpea plants.

### 2.7. Changes in Defense-Related Enzyme Activity

We further investigated the effects of HMSNs-406 on the activity of disease-related enzymes in cowpea roots to characterize the impacts of cowpea SAR. As shown in Figure 7, FOP infection significantly increased the POD activity by 2.22-fold in cowpea roots. The POD activity in the foliar HMSNs-406-treated cowpea roots was significantly decreased by 59.81% as compared with the infected control. Also, Si(OH)_4_ treatment led to a POD activity decline of 83.23%. This is likely because POD was reverted to the level of the healthy control after the suppression of pathogen activity and damage by HMSNs-406 and Si(OH)_4_ [29,31]. The change trends of the activity of the other two antioxidant enzymes, PAL and PPO (polyphenol oxidase), were similar to that of POD in FOP-infected roots, despite lesser difference magnitudes (Figure 7).

## 3. Discussion

HMSNs exert their Fusarium wilt suppression in cowpea plants in a size- and concentration-dependent manner. SiO_2_NPs have been a hot topic of recent interest as an emerging resistance inducer for plant disease protection [15,18,20]. However, their character differences in size, morphology, surface chemistry, etc., have resulted in differences of findings on the effects and mechanisms in alleviating pathogen stress [15,19,20]. Particle size, a core character determining many unique properties of nanoparticles, can affect the roles of SiO_2_NPs in plant disease control. However, scarce information is available due to differences of target pathogens and host plants used in previous studies [15]. SiO_2_NPs were proved to significantly reduce the Fusarium wilt severity in watermelon at a size range of 30–60 nm [9,10]. SiO_2_NPs with a size range of 5–100 nm were also reported to enhance disease resistance in many plants [15,21,32]. SiO_2_NPs with a size exceeding 100 nm, however, were rarely discussed, although they can also transfer in plants via leaves with a size up to 300 nm [33,34,35]. HMSNs are a unique class of SiO_2_NPs widely accepted as desirable pesticide carriers. It is also not known whether HMSNs can induce resistance in plants, and whether their performances will be size dependent. Thus, in this study, three sizes (19, 96, and 406 nm) of HMSNs were synthesized and characterized to determine their roles in suppressing FOP infection in cowpea roots by foliar application using phenotypes, fresh biomass, and disease progression as the indicators. The results showed that the effectiveness of HMSNs was size- and concentration-dependent, where the 406 nm-sized HMSNs (HMSNs-406) were the best at 1000 mg/L for cowpea Fusarium wilt control through foliar spray, with up to 40.00% reduced disease incidence (Figure 5a). The three HMSNs at a low concentration of 100 mg/L, as well as HMSNs-19 at a high concentration of 1000 mg/L, were observed to have little effect on alleviating disease incidence (Figure 5a). The fresh biomass of cowpea shoots and roots was also increased by HMSNs-406 (Figure 5c). The optimal concentration of HMSNs-406 of 1000 mg/L is within the expected range described in previous studies [7,9]. Interestingly, HMSNs of larger sizes exhibited a better capacity for Fusarium wilt suppression in cowpea plants. This seems to be contradictory to the general knowledge that smaller nanoparticles retain better efficacy [36,37]. El-Shetehy et al. (2021) proposed a mode of action of leaf-applied SiO_2_NPs: SiO_2_NPs can enter into the spongy mesophyll space via the stomata to activate plant immunity responses, probably by slow release of Si(OH)_4_, closure of the stomata, or interaction with their adjacent plant cells [7]. Thus, we can speculate that variances in both size-related dissolution rate of HMSNs in the leaf tissue and interactions of intact HMSNs with mesophyll cells might explain the size effects. The specific mechanisms need to be elucidated in follow-up studies.

Foliar spray of HMSNs-406 enhanced FOP resistance in cowpea roots via the SA-dependent SAR pathway. Currently, three modes of action (direct restraint [18], resistance induction [7,12,21] and cooperation [32]) have been involved in the functions of SiO_2_NPs on plant disease control. Induction of resistance is the most commonly reported mechanism of SiO_2_NPs [15] that was also demonstrated to be suitable for HMSNs in this study since HMSNs reduced the FOP severity in cowpea plants, but exhibited no direct toxic effect on FOP growth in vitro. Salicylic acid (SA) is a key plant hormone for plant immunity mediation including SAR. Importantly, the utilization of exogenous SiO_2_NPs could enhance SA levels to initiate SAR which suppressed a multitude of diseases in many plant species, such as Arabidopsis [7], peanut [21], tomato [22], and rice [12]. SiO_2_NPs rely on SA signaling to induce plant SAR to fend off pathogens by increasing the SA content [12,21] and consequently upregulating the expression of SA-responsive genes [7,12]. Our results also revealed that foliar spray of HMSNs-406, a unique type of SiO_2_NPs of 406 nm, enhanced cowpea resistance to FOP at 1000 mg/L through SA-activated SAR, showing both higher SA content (4.3-fold) and upregulation of SA marker genes of *PR-1* (1.97-fold) and *PR-5* (9.38-fold) in FOP infected cowpea roots relative to the infected controls (Figure 6). Although previous studies and our work indicate that SiO_2_NPs can trigger the production of SA to enhance plant resistance to pathogens, the specific mechanisms of how SiO_2_NPs trigger SA production in plants remain unknown. The accumulation of SA in plants can be regulated by genes involved in SA biosynthesis (*ICS*, *EDS5*, *PBS3*, *PAL*, *CYP73A*) and metabolism (*UGT*, *DMR6*, *SAMT*) [38,39]. Wu et al. (2023) revealed that SA biosynthesis is believed to occur via two routes of the ICS pathway (approximately 90%) and the PAL pathway (10%) [40]. The transcriptome results of a recent study indicated that SiO_2_NPs enhanced SA content in peanuts through the regulation of SA-metabolism genes *UGT*, *DMR6*, and *SAMT*, but the main gene *ICS* for SA biosynthesis did not respond to the SA changes. Our results showed that the expression of *PAL* significantly increased (8.54-fold) by HMSNs-406 foliar exposure. However, further studies will be required to determine whether other pathways may also participate in the HMSNs-406-induced SA changes in cowpea plants. El-Shetehy et al. (2021) found that SiO_2_NPs induced plant SAR between local leaves and systemic leaves in *A. thaliana* [7], while Du et al. (2022) described the roles of SiO_2_NPs on stimulating rice SAR, spreading from roots to leaves [12]. In this study, we further extended the SAR range, spreading from leaves to roots.

HMSNs have great potential to be developed as a new type of green pesticide for plant disease protection. Different kinds of nanoparticles have been found to have different uses. Metal-based nanoparticles such as silver (AgNPs), copper, and zinc nanoparticles have direct antibacterial or antifungal capability against a wide variety of plant pathogens [12,14,41,42]. SiO_2_NPs [7,12,20,21,22], as well as AgNPs [12,16] can function as resistance inducers to stimulate plant innate immunity, enhancing a plant’s resistance to many pathogens. Unlike the conventional SAR-inducing compounds such as benzothiadiazole that can reduce crop yields, SiO_2_NPs seem to have no negative effects on the growth and yield of plants. Of note, SiO_2_NPs, especially HMSNs, have been also widely used as nanocarriers of pesticides, nutrients, and biomacromolecules of functional proteins and nucleic acids, to improve their efficacy [25,26,27]. Recently, SiO_2_NPs with a specific mesoporous structure exhibited strong antimicrobial activity via directly inducing intracellular peroxidation damage of *Phytophthora infestans* [18]. Thus, SiO_2_NPs, especially HMSNs, have the greatest potential to be developed as a new versatile nano-fungicide relative to other nanoparticles. Previous studies, however, mainly focused on the exceptional properties of HMSNs as nanocarriers. In this study, for the first time, we demonstrated the potential of HMSNs as a nanoscale resistance inducer to enhance Fusarium wilt resistance in cowpea plants with the appropriate size and concentration. Moreover, our results also verified that HMSNs proved to be safer for plants compared with direct Si(OH)_4_ application since HMSNs had no adverse effects on cowpea seed germination (Figure 2) and growth (Figure 3) consistent with previous studies in other crops [7,9,12,21]. We also found that the mechanism of HMSNs was involved in activating SA-dependent SAR that can not only avoid either application of irreversible genetic modifications or fungicides of potential environmental risks [7], but also remove the space limitation of pesticide application. Combining our results and results of previous studies, a new type of versatile pesticide based on HMSNs can be developed to achieve the synergistic interactions of smart delivery, good antimicrobial effectiveness, and resistance induction for efficient plant disease control.

Based on the above discussions, a model of the HMSNs-initiated SA-dependent SAR pathway illustrating their antifungal mechanisms is presented in Figure 8.

## 4. Materials and Methods

### 4.1. Materials

Tetraethyl silicate (TEOS, 99%), cetyltrimethylammonium bromide (CTAB, 98%), Pluronic^®^F-108 (14,600 Da), homotrimethylbenzene (TMB, 98%), potassium silicate (Si(OH)_4_, 28%), and dimethyldimethoxysilane (MSDS, 98%) were purchased from Shanghai Aladdin Biochemical Technology Co., Ltd. (Shanghai, China). Anhydrous ethanol (99%), sodium carbonate (Na_2_CO_3_, 98%), hydrochloric acid (37%), and aqueous ammonia solution (25–28 wt%) were obtained from Sun Chemical Technology Co., Ltd. (Shanghai, China). Spray adjuvant Mairun was purchased from BeijingGrand AgroChem Co., Ltd. (Beijing, China). All other chemicals were commercially analytical grade products, unless otherwise specified. Dialysis bags (MWCO = 14,000 Da) were purchased from Beijing Solarbio Science and Technology Co., Ltd. (Beijing, China).

The target pathogenic fungus of *Fusarium oxysporum* f. sp. *phaseoli* (FOP) in this work was isolated from infected cowpea plant roots collected from the Ledong area of Hainan Province of China, and stored in our laboratory. This FOP strain was activated on PDA (potato dextrose agar) plates three times prior to use.

The cowpea cultivar Fengjiang1, widely cultivated in southern China, was selected as the target plant for the FOP suppression test. The potting soil mix was prepared by mixing a growth media of Pindstrup Plus with an equal weight of cultivated soil collected from the cowpea field.

### 4.2. Synthesis of Hollow Mesoporous Silica Nanoparticles (HMSNs)

The solid core silica nanospheres of 50 nm (sSiO_2_-50) were prepared by the existing methods [43,44]. For details, 10 mL of tetraethyl orthosilicate (TEOS) was added to a reaction mixture containing 20 mL of distilled water, 81 mL of anhydrous ethanol, and 2.93 mL of NH_3_-H_2_O (25–28%) at 70 °C. Hydrolysis and polycondensation of TEOS for 3 h yielded the target particles with the particle size of 50 nm. Then, the sSiO_2_-50 was separated by centrifugation (15,000× *g*, 15 min) and washed three times with distilled water. After the last washing, the system was purified by dialysis with distilled water, and the sSiO_2_-50 was obtained and directly used for the subsequent reactions.

Synthesis of the sSiO_2_-250 was carried out by varying the inputs of raw materials and the reaction time [43]: TEOS (3 mL) was rapidly added to a mixed system of ethanol (37 mL), distilled water (5 mL), and ammonia solution (25–28%, 1.6 mL). The system was stirred and reacted at room temperature for 1 h to produce a suspension of white colloidal silica nanoparticles. The sSiO_2_-250 was separated by centrifugation and washed with distilled water and ethanol, and the target sSiO_2_-250 was obtained and directly used for the subsequent reactions.

The 96 and 406 nm-sized HMSNs (HMSNs-96 and HMSNs-406) were synthesized according to previously describe methods [43,44] using sSiO_2_-50 and sSiO_2_-250 as the cores, respectively. In detail, sSiO_2_ (200 mg) was ultrasonically dispersed in 40 mL of distilled water, then the system was added into a mixture of CTAB (300 mg), distilled water (60 mL), ethanol (60 mL), and ammonia solution (25–28%, 1.1 mL) at room temperature. The mixture was stirred for 0.5 h, then 0.5 mL of TEOS was added to the system and the reaction was continued for another 6 h. After the reaction, the solid particles were centrifuged and re-dispersed by ultrasonication in 40 mL of distilled water, and 848 mg of Na_2_CO_3_ was added to the system and the reaction was stirred for another 10 h at 50 °C. The obtained particles were then washed with ethanol and water, and dried. The resulting solid was placed in a muffle furnace and heated to 550 °C at a ramping rate of 1.5 °C/min, then kept for 6 h to remove the CTAB surfactant to obtain the target HMSNs-96 and HMSNs-406.

HMNSs-19 were prepared according to a previously reported method [45]. Pluronic^®^F-108 (1 g) and HCl (5 mL) were dissolved in distilled water (25 mL) and stirred vigorously to form a homogeneous system. After Pluronic^®^F-108 was completely dissolved, homotrimethylbenzene (0.8 g) was added to the above system and stirred continuously (1000 rpm) at 25 °C for 3 h. Then, TEOS (1.0 g) was added dropwise to the reaction system at a rate of 0.1 mL/min under stirring. After the reaction was continued for 5 h, dimethyldimethoxysilane (0.4 g) was added dropwise at a rate of 0.1 mL/min, and the reaction was continued for another 36 h. The obtained milky-white system was dialyzed for 48 h to remove the hydrochloric acid, then lyophilized in a vacuum lyophilizer. The resulting solid pellet was dispersed in 30 mL of ethanol–HCl (*v*/*v*, 29:1) mixture and the system was refluxed for 12 h to remove the surfactant, and the process was repeated twice. Afterwards, the solid product was collected by centrifugation (10,000 rpm, 15 min) and washed repeatedly with ethanol to completely remove the Pluronic^®^F-108 and HCl. Finally, the solid particles after template removal were dried under vacuum at 60 °C overnight to obtain the target HMNSs-19.

### 4.3. Electron Microscopy Observation

Prior to TEM observation, each HMSNs sample was diluted to 1 mg/mL in ethanol and sonicated for 15 min to ensure dispersity. Afterward, 2 μL of the suspension was dropped onto a carbon-coated copper grid (200 mesh, Beijing Zhongjingkeyi Technology Co., Ltd., Beijing, China) and air-dried overnight. The morphologies of the three HMSNs were then observed with a JEOL JEM-2100 instrument (Tokyo, Japan) at an accelerating voltage of 120 kV. To determine the size of the HMSNs, the images were analyzed using ImageJ [46] (https://imagej.nih.gov/ij/, accessed on 10 November 2023) and the diameters of at least 500 randomly selected nanoparticles were measured.

For the scanning electron microscope (SEM) observation, 2 μL of the HMSNs suspension was dropped onto a piece of monocrystalline silicon (0.5 cm × 0.5 cm) and air-dried overnight. The samples were then gold-coated using an E1010 sputter coating machine (Hitachi, Tokyo, Japan) for 60 s, and imaged using a JSM-6360LV SEM (JEOL, Tokyo, Japan).

### 4.4. Dynamic Light Scattering (DLS) and Zeta Potential Measurements

To measure the hydrodynamic diameters and surface potentials of the three HMSNs, each nanoparticle sample was suspended in water at 0.5 mg/mL and sonicated for 15 min to ensure uniform dispersion. A Malvern Instrument of Zetasizer Nano ZS90 (Worcestershire, UK) was then used for DLS and zeta potential measurements.

### 4.5. Monitoring Seed Germination and Seedling Growth after HMSNs Treatment

Cowpea seeds were soaked into a 75% ethanol–water solution (*v*/*v*) for 5 min surface sterilization and washed with sterile water fully to remove the surface alcohol. Then, six of the sterilized seeds were set into a 9 cm petri dish with a piece of filter paper (9 cm in diameter) immersed in 10 mL of HMSNs suspension at a given concentration of 100, 500, 1000, or 1500 mg/L. Sterile water was used as the blank control. Each treatment was repeated ten times. All the cowpea seeds were incubated under identical conditions (28 ± 1 °C, 75 ± 10% relative humidity, and a12:12 light/dark photoperiod). Seeds began to sprout at 12 h after treatment and were counted every 2 h for 24 h, during which over 80% of seeds sprouted in the control groups. The germination index (GI) of cowpea was calculated by the Equation (1) [47] as followed:Germination index = ∑G_t_/D_t_(1)
where G_t_ represents the number of germinations on the hour, and D_t_ represents the number of germination hour.

The root and shoot fresh biomass of seedlings were also measured after 7 days of treatment.

### 4.6. Mycelial Growth Inhibition Test

The direct suppression of the three HMSNs against *F. oxysporum* f. sp. *phaseoli* (FOP) growth was determined by the previously reported agar dilution method [48] using the mycelial growth inhibition as the indicator. Briefly, stocks of the three HMSNs were prepared by ultrasonic dispersion in sterile water to gain the final concentrations of 1, 10, and 20 mg/mL, respectively. Each stock solution (5 mL) was added to molten PDA (50 mL) at a temperature below 50 °C. After sufficient mixing, the mixture was poured immediately into 90 mm Petri dishes to form plates of 2–3 mm thickness with final HMSNs concentrations of 100, 1000, and 2000 mg/L. PDA plates supplemented with sterile water served as the controls. Afterwards, a 5 mm diameter mycelial disc from the actively growing colony front of FOP was then placed in the center of each plate with the inoculum side down. All the plates were then incubated in the dark at a constant temperature of 25 °C for 7 days. Each treatment was performed in triplicate. The mycelial growth diameters were measured and subsequently converted into the inhibition rates according to the following Equation (2):Inhibition rate (%) = [(D_c_ − D_t_)/(D_c_ − 0.5)] × 100(2)
where D_c_ and D_t_ indicate the mycelial growth diameters of the controls and treatment groups, respectively.

### 4.7. Pot Experiments

Pot experiments were carried out according to the previously described processes [9,29]. FOP was inoculated into potato dextrose liquid medium (PDB) for 7 days at 25 °C and 120 rpm. Afterwards, FOP conidia were harvested by filtration through three layers of sterile gauze followed by centrifugation (3000× *g*, 5 min). The precipitated conidia were re-suspended in sterile water and diluted to 1 × 10^6^ conidia/mL. Then, the conidia suspension was mixed with the potting soil mix to prepare the infected soil (1 × 10^6^ conidia/g dry soil). The potting soil mix supplied with sterile water was used as the noninfected soil.

Cowpea seeds were soaked in sterile water for 24 h for pre-germination, then planted in the noninfected soils. When the plants reached the three- to four-leaf stage, seedlings of uniform size were transplanted into the infected soil for foliar nanoparticle exposure. The HMSNs dispersions for foliar spray were prepared by adding 0.1% of Pluronic^®^F-108 and 0.1% of Mairun as dispersive adhesives to make the final concentrations of 100 and 1000 mg/L, respectively. A volume of 5 mL of HMSN dispersion was sprayed on the leaf surface of each seedling, and this treatment was repeated twice at 5 d intervals after the first application. The infected controls, as well as the healthy controls in the uninfected soil, were similarly established by spraying with distilled water supplied with the same amounts of dispersive adhesives. Foliarly spraying a potassium silicate solution of identical concentration to the HMSNs on the seedlings in the infected soil served as the Si(OH)_4_ treatments. Each treatment contained 12 seedlings and was repeated three times. All the pots were set in a greenhouse for the duration of the study.

The cowpea plants were evaluated for the severity of Fusarium wilt at 14, 21, and 29 days of post-transplanting in the infected soil, using a 0 to 4 scale monitoring both the external leaf and internal vascular symptoms modified from previous studies [49,50]. The detailed grading standards can be seen in the Appendix A. The disease progress on the cowpea plants was exhibited by the area under the disease progress curve (AUDPC) of the cumulative severity ratings plotted as a function of time [10,51]. AUDPC was calculated using the previously described trapezoid rule (Equation (3)) [10,51]:(3)$$\mathrm{AUDPC}=\frac{\sum \left[y_{i}+y_{\left(i+1\right)}\right]}{2\times \left[t_{\left(i+1\right)}-t_{i}\right]}$$
where, in this equation, *y_i_* and *y*_(*i*+1)_ are the disease severity ratings at the time of *t_i_* and its adjacent time interval *t*_(*i*+1)_ for rating, respectively.

After 29 days, the pot experiments were terminated and the root and shoot fresh biomass were determined.

### 4.8. Salicylic Acid (SA) Measurement

Cowpea plants were exposed to 1000 mg/L of HMSNs-406 by foliar application according to the procedure as described for the pot experiments. Potassium silicate and water were used as the Si(OH)_4_ control and blank control, respectively. The plant roots of different treatments were collected at 29 days of post-transplanting in the infected soil and stored in triplicate at −80 °C for subsequent tests of salicylic acid contents, RNA extraction, and enzyme activity analysis. Quantitative determination of overall SA of free and conjugated forms in cowpea roots was performed according to a previously described ultra-performance liquid chromatography (UPLC) method [52,53] using a Rigol L3000 UPLC (Beijing, China) with fluorescence detection.

A weight of 0.2 g of each fresh root sample was ground under liquid nitrogen to obtain the root homogenate. After homogenization, 1 mL of a precooled methanol aqueous solution (*v*:*v*, 9:1) was added to extract total SA at 4 °C overnight. The mixture was then centrifuged (10 min, 8000× *g*) to obtain the supernatant and precipitate. The precipitate was extracted again using 0.5 mL of the precooled methanol aqueous solution. After centrifugation, the second supernatant was obtained and merged with the first supernatant. The total supernatant was concentrated by the vacuum evaporation method at 40 °C to remove methanol. After that, 20 μL of a trichloroacetic acid aqueous solution (1 mg/mL) was added into the above concentrate. After sufficient mixing by shaking, 1 mL of an ethyl acetate–cyclohexane mixture (*v*:*v*, 1:1) was used to extract the free SA twice. The total organic phase was then dried with nitrogen blowing and dissolved by methanol for UPLC analysis. The remaining water phase was mixed with 0.5 mL of a HCl solution (2 mol/L) and the conjugated SA in the system was hydrolyzed at 80 °C for 1 h to become free form. Then, the SA was extracted with 1 mL of an ethyl acetate–cyclohexane mixture (*v*:*v*, 1:1) twice. The obtained organic phase was then dried with nitrogen blowing and dissolved by methanol for UPLC analysis.

Quantitative determination of overall SA of free and conjugated forms in cowpea roots was performed on a Rigol L3000 UPLC (Beijing, China) with fluorescence detection. A Compass-C_18_ column (250 mm × 4.6 mm, 5 μm particle size) was used for SA separation at 35 °C, using a methanol–water mixture (3:2, *v*/*v*) supplemented with 0.4% acetic acid as the elution. The injection volume was 10 μL and the flow rate was 0.8 mL/min. The excitation wavelength was set as 294 nm and the fluorescence signal was collected at 426 nm. A SA standard curve was established using this method for quantitative determination of SA in samples. Total SA was calculated by the sum of free and conjugated SA. Each test was repeated three times.

### 4.9. Gene Expression Variations

A quantitative real-time PCR (qRT-PCR) assay was employed to monitor changes in the expression of select resistance-related genes after HMSN treatment. Cowpea roots collected for SA measurement were also used for total RNA extraction to observe resistance-related gene expression variations. After being grounded in liquid nitrogen, 150 mg of plant tissue homogenates were collected for total RNA extraction using a TaKaRa MiniBEST Plant RNA Extraction Kit (Dalian, China). The concentration and quality of the extracted RNA samples were assessed using a Thermo Scientific Nanodrop Lite Spectrophotometer (Wilmington, DE, USA). The cDNA was synthesized from the extracted RNA samples using a QuantiTect Reverse Transcription kit (Qiagen, Hilden, Germany). The eukaryotic elongation factor gene of *EF1b* was used as the internal control. The optimal primers for gene amplification and PCR programs are presented in Appendix A, respectively. qRT-PCR was then performed on a Bio-Rad CFX96 Touch Real-Time PCR Detection System (Hercules, CA, USA) using SYBR Green as the fluorescent intercalating dye. The relative expression of genes was calculated by the 2^−ΔΔCt^ method.

### 4.10. Defense-Responsive Enzyme Activity Measurement

Three defense-responsive enzymes of phenylalanine ammonia lyase (PAL), peroxidase (POD), and polyphenol oxidase (PPO) were selected as the target enzymes, and their activities were determined according to the instructions of relevant commercial assay kits from Beijing Solarbio Science and Technology Co., Ltd. (Beijing, China). Three replicates were performed for each experiment.

### 4.11. Statistical Analysis

Data are represented as mean ± standard deviation. In the experiments involving cowpea plants, one-way ANOVA was carried out by SPSS 23.0 software (SPSS Inc., Chicago, IL, USA) to test for statistical significance (*p* < 0.05), comparing the response of plants treated with different HMSNs.

## 5. Conclusions

The present study indicated that foliar spray of HMSNs enhanced cowpea resistance to FOP in a size- and concentration-dependent manner. HMSNs-406, with an average size of 406 nm, possessed the best control effect at a concentration of 1000 mg/L with an up to 40.00% decline in the disease severity. The three HMSNs at a low concentration of 100 mg/L, as well as HMSNs-19 at a high concentration of 1000 mg/L, were observed to have little effect on alleviating the disease incidence. Foliar spray of HMSNs-406 also promoted the growth of FOP-inoculated cowpea plants. Moreover, foliar application of HMSNs-406 (1000 mg/L) increased the salicylic acid (SA) content in cowpea roots, as well as the expression levels of SA marker genes *PR-1* and *PR-5*, and its receptor gene *NPR-1*. Overall, HMSNs-406 exerted their FOP suppression in cowpea plants via activating SA-dependent SAR responses rather than directly suppressing FOP growth. As a desirable carrier extensively discussed, our results extend a new role for HMSNs as resistance inducers without adverse effects on plant growth, to be used as a low-cost, highly efficient, safe and sustainable alternative for plant disease protection.

## Figures and Tables

**Figure 1 ijms-25-04514-f001:**
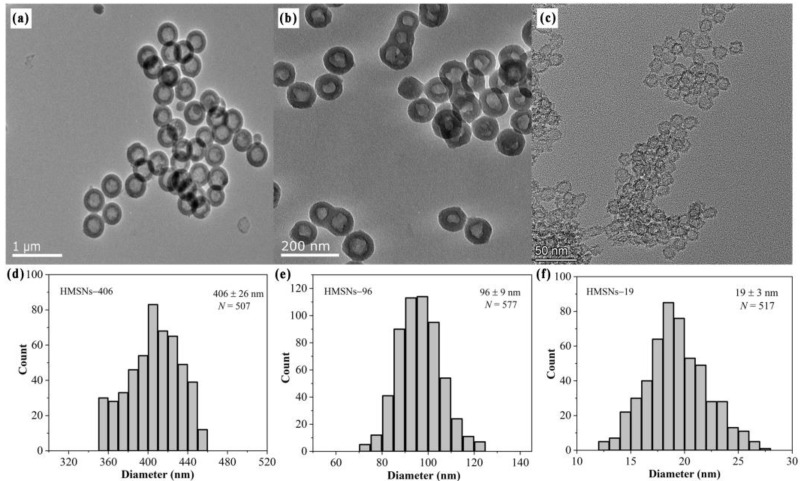
HMSNs under investigation. TEM images of (**a**) HMSNs-406, scale bar: 1 µm; (**b**) HMSNs-96, scale bar: 200 nm; and (**c**) HMSNs-19, scale bar: 500 nm. Particle size distributions of (**d**) HMSNs-406, (**e**) HMSNs-96, and (**f**) HMSNs-19 based on the TEM image analysis. Averages ± standard deviations.

**Figure 2 ijms-25-04514-f002:**
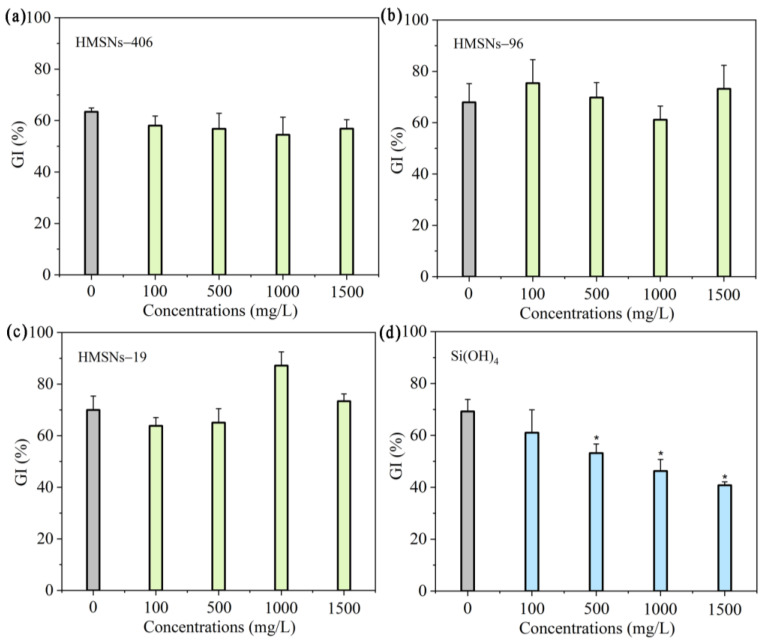
Effect of HMSNs (**a**–**c**) and Si(OH)_4_ (**d**) on germination of cowpea seeds. Concentrations were set as 100, 500, 1000, and 1500 mg/L, respectively. Error bars are averages and standard deviations of ten replicates. Asterisks (*) represent significant differences as compared with the sterile water controls (0 mg/L) using one-way ANOVA mode for significance testing with Dunnett’s multiple comparisons test at *p* < 0.05.

**Figure 3 ijms-25-04514-f003:**
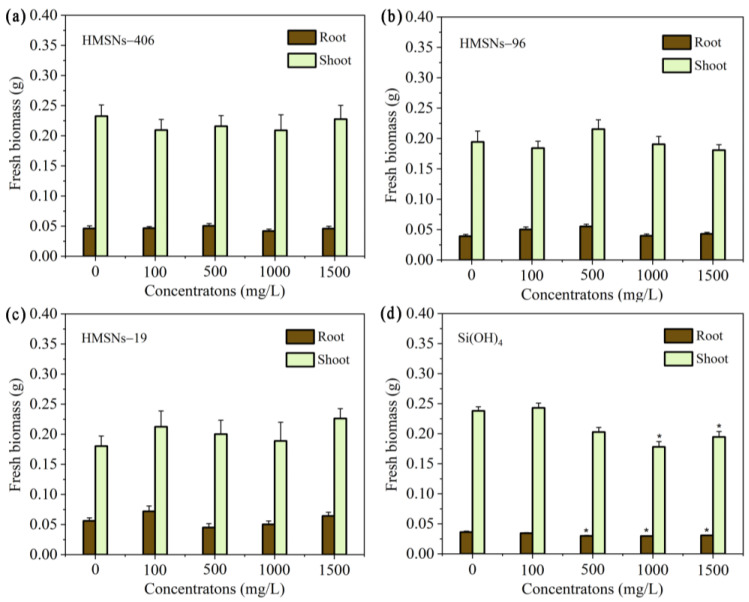
Effect of HMSNs (**a**–**c**) and Si(OH)_4_ (**d**) on cowpea seedling growth. Concentrations were set as 100, 500, 1000 and 1500 mg/L, respectively. Error bars are averages and standard deviations of ten replicates. Asterisks (*) represent significant differences as compared with the sterile water controls (0 mg/L) using one-way ANOVA mode for significance testing with Dunnett’s multiple comparisons test at *p* < 0.05.

**Figure 4 ijms-25-04514-f004:**
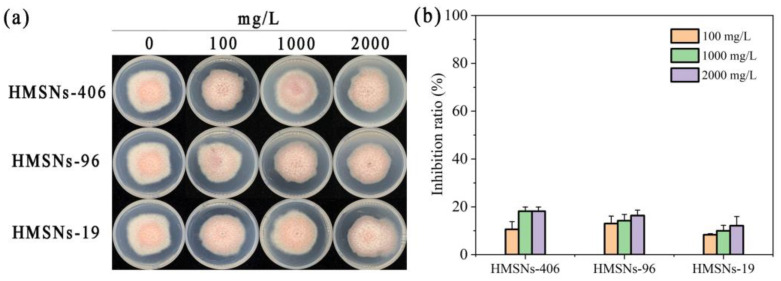
Effect of three HMSNs on the mycelial growth of *Fusarium oxysporum* f. sp. *phaseoli* (FOP) in vitro. (**a**) Mycelial growth phenotypes of FOP treated with three HMSNs at concentrations ranging from 100 to 2000 mg/L. (**b**) Inhibition ratios of three HMSNs against FOP mycelial growth at different concentrations of 100, 1000, and 2000 mg/L.

**Figure 5 ijms-25-04514-f005:**
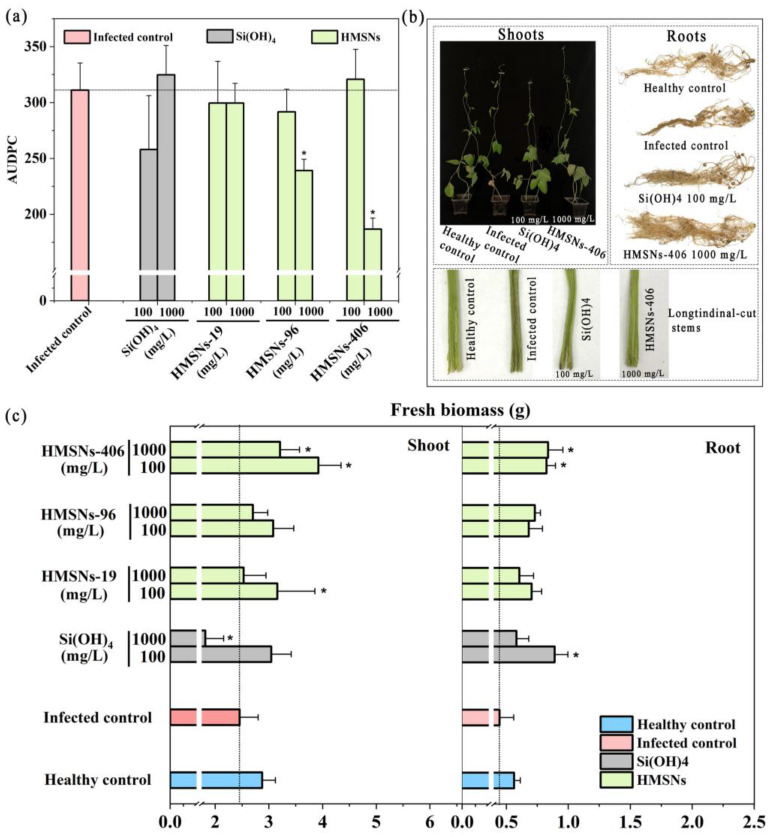
HMSNs enhance the disease resistance in a size- and concentration-dependent manner. (**a**) Effects of HMSNs and Si(OH)_4_ on the cowpea Fusarium wilt incidence by foliar application using AUDPC as the indicator. (**b**) Phenotypic images of cowpea shoots, longitudinal-cut stems and intact roots by HMSNs-406 (1000 mg/L) or Si(OH)_4_ (100 mg/L) foliar treatment for 29 days. (**c**) Changes in the shoot and root biomass of cowpea foliarly treated with HMSNs or Si(OH)_4_ at different concentrations of 100 and 1000 mg/L for 29 days. The healthy control represents cowpea plants growing in the noninfected soil and treated with water. Other treatments represent cowpea plants growing in the infected soil and foliarly treated with water (the infected control), Si(OH)_4_, or HMSNs-406. Error bars are averages and standard deviations of three replicates. Asterisks (*) in (**a**,**c**) represent significant differences as compared with the infected controls using one-way ANOVA mode for significance testing with Dunnett’s multiple comparisons test at *p* < 0.05.

**Figure 6 ijms-25-04514-f006:**
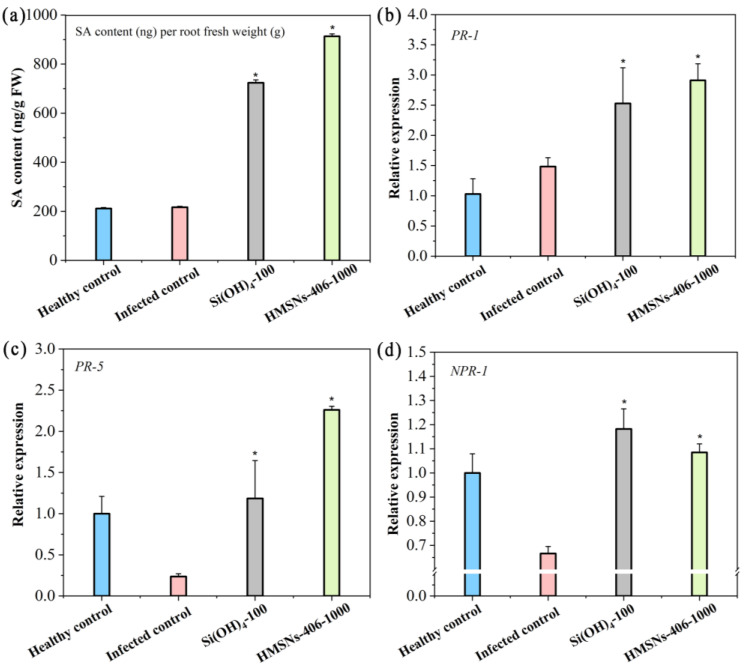
HMSNs-406 induce Fusarium wilt resistance based on SA-dependent SAR pathway in cowpea. (**a**) Changes in salicylic acid (SA) content in cowpea roots. RT-qPCR analysis of the gene expression of the SAR-related genes *PR-1* (**b**), *PR-5* (**c**), and *NPR-1* (**d**) in cowpea roots. Root samples were collected 29 days after FOP infection. *EF1b* was used as the reference gene. The healthy control represents cowpea plants growing in the noninfected soil and treated with water. Other treatments represent cowpea plants growing in the infected soil and foliarly treated with water (the infected control), Si(OH)_4_ (100 mg/L) or HMSNs-406 (1000 mg/L). Error bars are averages and standard deviations of three replicates. Asterisks (*) represent significant differences as compared with the infected controls using one-way ANOVA mode for significance testing with Dunnett’s multiple comparisons test at *p* < 0.05.

**Figure 7 ijms-25-04514-f007:**
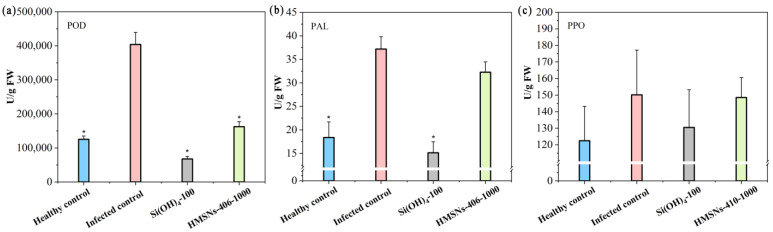
HMSNs-406-induced changes in the activity of disease-related enzymes of (**a**) peroxidase (POD), (**b**) phenylalanine ammonia lyase (PAL), and (**c**) polyphenol oxidase (PPO) in cowpea roots infected by FOP. Root samples were collected 29 days after FOP infection. The healthy control represents cowpea plants growing in the noninfected soil and foliarly treated with water. Other treatments represent cowpea plants growing in the infected soil and foliarly treated with water (the infected control), Si(OH)_4_ (100 mg/L) or HMSNs-406 (1000 mg/L). Error bars are averages and standard deviations of three replicates. Asterisks (*) represent significant differences as compared with the infected controls using one-way ANOVA mode for significance testing with Dunnett’s multiple comparisons test at *p* < 0.05.

**Figure 8 ijms-25-04514-f008:**
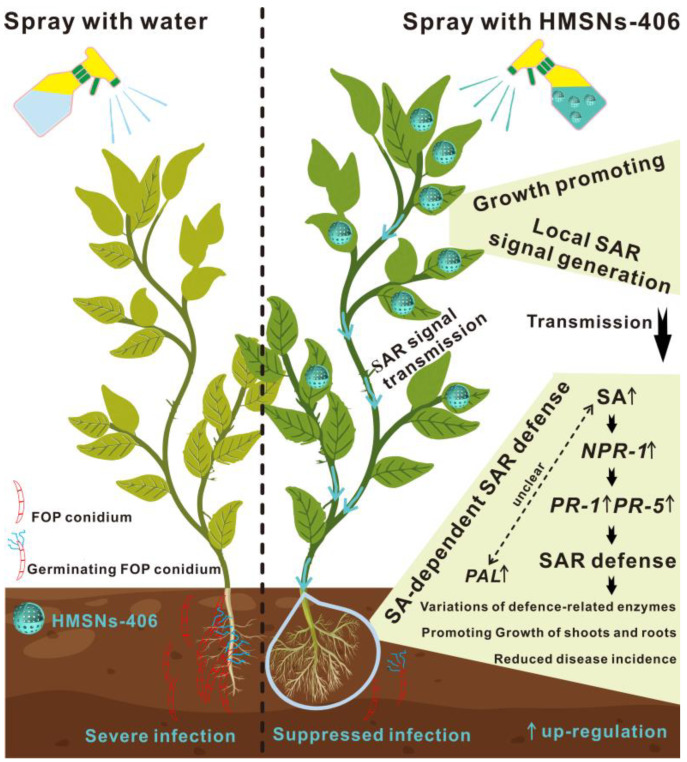
Foliar spray of HMSNs-406 enhanced FOP resistance in cowpea roots via the SA-dependent SAR defense pathway.

## Data Availability

The datasets generated and analyzed during the current study are available from the corresponding authors upon reasonable request.

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
