# Peer review of "Hollow Mesoporous Silica Nanoparticles as a New Nanoscale Resistance Inducer for Fusarium Wilt Control: Size Effects and Mechanism of Action"

_ijms, 2024, doi:10.3390/ijms25084514_

Round 1

Reviewer 1 Report

Comments and Suggestions for Authors

Dear authors,

The subject addressed is an interesting one. However, the article is sometimes not clear enough, especially regarding the working method and the interpretation of the results.

Here are our observations.

Abstract section. Please re-read the text, lines 28-32, and rewrite it for greater clarity.

Results section. Lines 156-164.

Since in the article there is a subsection "2.3. Toxicity of HMSNs to Fusarium oxysporum f. sp. phaseolus (FOP) in vitro" related to the toxicity of HMSN compounds - the authors must insert Figure S1b in the article - otherwise the readers would be confused and with an image unclear on their effect.

Are the chemicals used elicitors in the proper sense of the word? Argue this in the Discussion section through detailed comparisons with other compounds described in the literature that function as elicitors.

The existence of a high content of salicylic acid may be due to external factors rather than the action of the compounds used in the experiments. What would be the mechanism of action of HMSN that triggers the production of salicylic acid and consequently increases the resistance of plants to the pathogen? Please explain in the Results Section and elaborate in the Discussions section by including more articles on this topic to justify your statements.

Materials and methods section. The lines - explain the changes to the original method, the methods must be described, even in a shortened form so that researchers interested in the topic can reproduce the respective experiments. The supplementary data sets that support the Results should be found in Supplementary Materials.

The same observation for the procedures cited in lines 381-384; 417-418; 461-463.

Conclusions section. Unclear, confused sentences and phrases.

Line 491 - "we conformed the role" is not correct. You probably meant to say confirmed, The same mistake can be found in the Introduction section.

Lines 493-494 The phrase is completely unclear. What did you mean? Please rephrase.

The entire Conclusions section must be reformulated linguistically correctly and to highlight the original argument of the authors. Also, please explain the meaning in which the word "elicitor" is used in the DISCUSSION section, what is its definition and to what extent it fits with with the compounds used in your experiments.

With best regards!

Comments on the Quality of English Language

Minor error typos detected.

Author Response

Dear Reviewer,

Thank you for your letter and for your comments with regard to our manuscript (ID: ijms-2913222). We have carefully studied the valuable comments and tried our best to revise and improve the manuscript according to the helpful comments.

All the required experimental data have been supplied in the revised manuscript and other necessary revisions have been done and marked in red in our manuscript.

Attached please find a combined document of responses to the comments, the revised manuscript with the revised contents labeled by red color, and the revised Supplementary Materials.

Best wishes!

Sincerely,

Dr. Shujing Zhang on behalf of the authors.

Shujing Zhang, Ph. D. (Corresponding Author)

School of Tropical Agriculture and Forestry

Hainan University

Haikou, Hainan, 570228

P. R. China

E-mail: sjzhang@hainanu.edu.cn

Reviewer 2 Report

Comments and Suggestions for Authors

Title: Size-Dependent Disease Resistance Enhancement of Hollow Mesoporous Silica Nanoparticles in Cowpea Plant Involved in Salicylic Acid Mediated Systemic Acquired Resistance for Fusarium Wilt Control

This manuscript by Ding et al. evaluated the synthesized three sizes of HMSNs (19, 96 and 406 nm) silica nanoparticles influence on seed germination, seedling growth, and Fusarium suppression in cowpea roots. Overall, the manuscript is interesting, but the author should require revision as follows:

Comments

1.     The title should be crispier. Please revise.

2.     The abstract is weak, it can be more precisely presented.

3.     In general silica particles are highly biocompatible. Lines 67-73, many types of nanoparticles have been reported for the antimicrobial properties against plant pathogens i.e. silver and its composites. How are silica-based nanoparticles better, and are their mechanism of antimicrobial properties to be presented and compared here briefly with additional information? (for example, https://doi.org/10.1016/j.scitotenv.2023.168318)

4.     Provide SEM images of synthesized nanoparticles.

5.     Please provide one illustration to summarize the present study and highlight the mechanism of action of silica nanoparticles.

6.     The discussion can be minor polished with recent citations.

7.     Figures quality can be improved.

Comments on the Quality of English Language

Minor changes are required.

Author Response

Dear Reviewer,

Thank you for your letter and for your comments with regard to our manuscript (ID: ijms-2913222). We have carefully studied the valuable comments and tried our best to revise and improve the manuscript according to the helpful comments.

All the required experimental data have been supplied in the revised manuscript and other necessary revisions have been done and marked in red in our manuscript.

Attached please find a combined document of responses to the comments, the revised manuscript with the revised contents labeled with red color, and the revised Supplementary Materials.

Best wishes!

Sincerely,

Dr. Shujing Zhang on behalf of the authors.

Shujing Zhang, Ph. D. (Corresponding Author)

School of Tropical Agriculture and Forestry

Hainan University

Haikou, Hainan, 570228

P. R. China

E-mail: sjzhang@hainanu.edu.cn

Round 2

Reviewer 1 Report

Comments and Suggestions for Authors

Dear authors,

Thank you for the effort you have made. You brought all the necessary clarifications, so that the article is much improved compared to the original version and can be published.

With best regards!

Author Response

Dear Reviewer,

Thank you for your letter with regard to our manuscript (ID: ijms-2913222). We really appreciate your hard work for reviewing our manuscript and providing valuable comments to improve the manuscript. Thank you once again for your approvel of our work for publication.

Best wishes!

Sincerely,

Dr. Shujing Zhang on behalf of the authors.

Shujing Zhang, Ph. D. (Corresponding Author)

School of Tropical Agriculture and Forestry

Hainan University

Haikou, Hainan, 570228

P.R. China

E-mail: sjzhang@hainanu.edu.cn